# Therapeutic Strategies for Patients with Advanced Small Bowel Adenocarcinoma: Current Knowledge and Perspectives

**DOI:** 10.3390/cancers14051137

**Published:** 2022-02-23

**Authors:** Emilie Moati, Michael J. Overman, Aziz Zaanan

**Affiliations:** 1Department of Gastroenterology and Digestive Oncology, Institut du Cancer Paris Carpem, Assistance Publique des Hôpitaux de Paris, European Georges Pompidou Hospital, University of Paris, 75015 Paris, France; emilie.moati@aphp.fr; 2Department of Gastrointestinal Medical Oncology, MD Anderson Cancer Center, Houston, TX 77030, USA; moverman@mdanderson.org; 3Centre de Recherche des Cordeliers, INSERM UMRS1138, CNRS, Sorbonne Université, USPC, Université de Paris, Equipe Labellisée Ligue Nationale Contre le Cancer, CNRS SNC 5096, 75006 Paris, France

**Keywords:** small bowel adenocarcinoma, chemotherapy, targeted therapy

## Abstract

**Simple Summary:**

Small bowel adenocarcinomas (SBA) are rare tumors with a poor prognosis. Due to the rareness of this illness, there is limited scientific data to guide therapeutic management. The recent large prospective ARCAD-NADEGE study evaluated 347 patients with SBA and has helped to improve our current knowledge of this disease. When diagnosed with advanced, non-surgically resectable disease, chemotherapy remains the cornerstone of the treatment and appears to add a survival benefit compared to palliative care. Other more recent drugs, such as small molecule targeted therapeutic inhibitors or immunotherapy, may have a role in improving the outcome of patients with advanced SBA. In this review, we summarized the classical therapeutic arsenal as well as possible future promising treatments in advanced SBA.

**Abstract:**

Small bowel adenocarcinoma (SBA) is diagnosed at an advanced (unresectable or metastatic) tumor stage in approximately one-third of cases. This is partly due to the non-specific symptomatology and limitations in endoscopic and radiologic detection methods. In this context, the prognosis remains poor and systemic chemotherapy appears to benefit patients when compared to best supportive care alone, despite the absence of randomized controlled trials. The results of a recent large prospective cohort (ARCAD-NADEGE) reported that the absence of chemotherapy was a predictive factor for a lower overall survival (OS) even though poor differentiation and SBA associated with Crohn’s disease correlate with poor prognosis. In retrospective series, the median OS ranges from approximately 9 to 18 months with current treatment approaches. A combination of a fluoropyrimidine and oxaliplatin (FOLFOX or CAPOX) appears to be the most utilized and effective first-line chemotherapy regimen. Other front-line alternatives are the combination of 5-FU and cisplatin or fluoropyrimidine and irinotecan (FOLFIRI). In second-line, FOLFIRI is an effective option after progression on platinum-based therapy. Taxane-based therapy appears to be an alternative option, but further evaluation in larger series is needed. To a limited extent, the role of surgical resection for metastatic disease appears to be a valid option, though this approach has not been evaluated in prospective clinical studies. Due to the rareness of the disease, inclusion in clinical trials should be prioritized, and there is hope that targeted therapies and immunotherapy may enter the therapeutic arsenal for these patients.

## 1. Introduction

Small bowel adenocarcinoma (SBA), duodenal, jejunal, or ileum localizations, is a rare disease and accounts for 3–5% of gastrointestinal cancers. These tumors are diagnosed at an advanced (unresectable or metastatic) stage in around one-third of cases, in part due to the non-specific nature of clinical symptoms. With the development of improved imaging techniques and advances in endoscopy, such as enteroscopy and capsule endoscopy, it is possible that trends toward earlier diagnosis can be seen in the future [1,2,3]. In the NADEGE cohort study, the tumor stage at diagnosis was localized or resectable in 54%, locally advanced, unresectable in 5.5%, metastatic in 33.5%, and undetermined clinical stage in 7% [4]. As in many other cancers, the most important prognostic factor in SBA is the TNM classification. Despite a relative improvement in the median overall survival (OS) for localized disease, the overall prognosis for SBA remains poor. The 5-year overall survival (OS) rate is 50% for stage I, 40% for stage II, 10–40% for stage III, and reaching less than 5% for stage IV disease [5]. The other factors associated with a worse prognosis are male gender, duodenal location, poor differentiation, and SBA associated with Crohn’s disease as compared to de novo SBA [5,6,7,8]. Retrospective studies demonstrated the ability to resect limited metastatic disease, but no prospective studies were conducted. For unresectable metastatic disease, resection of the primary tumor should only be considered in the case of primary tumor symptomatology, such as perforation, bowel obstruction, or uncontrolled gastrointestinal bleeding. For all other situations, the main modality of therapy for metastatic SBA is palliative chemotherapy. This review focuses upon the role of systemic therapy, surgical metastasectomy, and the novel therapies such as targeted therapies and immunotherapy in advanced SBA.

## 2. Systemic Chemotherapy 

Although no randomized clinical trials have yet reported a benefit for systemic chemotherapy compared to best supportive care alone, retrospective studies suggested a benefit for palliative chemotherapy. One of the earliest retrospective studies reported overall survival (OS) of 12 versus 2 months for systemic chemotherapy and best supportive care, respectively [6]. Several retrospective or prospective studies since reported a survival benefit for the administration of chemotherapy compared to best supportive care alone, with a median OS ranging from 9 to 19 months in patients with palliative chemotherapy, whereas it ranged only from 2 to 13 months in patients with palliative care [4,6,9,10,11,12,13,14] (Table 1).

The results of a recent large prospective cohort (NADEGE) reported that the only predictive factor of a lower OS was the absence of chemotherapy. These data are, however, subjected to bias as patients who did not receive chemotherapy were more likely to be frail and have a poorer performance status [4]. In this observational study, a benefit of palliative chemotherapy was described with a median OS of 14.3 months for the 86 patients receiving systemic chemotherapy versus 2.2 months for 15 patients treated with palliative care only. Among all patients with metastatic disease, the median OS was 12.7 months [4]. Furthermore, although having a prognostic impact, the tumor location had no impact on the response to chemotherapy.

These survival results are consistent with those reported in several phase II studies with an OS ranging from 8 to 20 months and a PFS ranging from 3 to 11 months for patients treated with various chemotherapy regimens [15,16,17,18,19,20] (Table 2).

### 2.1. First-Line Chemotherapy Regimen 

In most studies: first-line chemotherapy was mainly 5-FU based regimens, with or without a platinum salt or irinotecan. These different regimens are shown in Table 1 and Table 2 [15,16,17,18,20,21,22,23,24,26,27,28].

In the NADEGE cohort, chemotherapy was given to 122 metastatic SBA patients with regimes being FOLFOX or capecitabine and oxaliplatin (CAPOX) in 69.8% of patients, FOLFIRI in 18.6% of patients, and 5-FU monotherapy in 7.0% of patients [4]. In summary, a fluoropyrimidine and oxaliplatin (FOLFOX or CAPOX) appear to be the most used and effective first-line chemotherapy.

One of the first studies of chemotherapy in SBA involving eight advanced SBA patients reported a median OS of 13 months and a median PFS of 7.8 months for patients treated with 5-FU-based chemotherapy. A second early study of 20 patients reported a median OS of 14 months for patients treated with the combination of 5-FU and a platinum salt [21,22].

The benefit of 5-FU combined with a platinum salt in comparison to other chemotherapy regimens is suggested by one retrospective study involving 80 patients with advanced SBA. In this series, median PFS and OS were 8.7 and 14.8 months, respectively, for 5-FU and cisplatin, which was significantly better than other chemotherapy regimens [23].

In a retrospective Japanese single-center study involving 132 patients, Tsushima et al. reported that FOLFOX demonstrated improved PFS and a trend towards better OS when compared to other chemotherapy regimens [26]. The potential benefit for a fluoropyrimidine and a platinum agent over other chemotherapy is supported by a retrospective multicenter study reported by Zaanan et al. In this study involving 93 SBA patients, FOLFOX treated patients had an improved OS of 17.8 months in comparison to other chemotherapy regimens such as 5-FU alone or combined with cisplatin or irinotecan [24]. In the subgroup of patients treated with platinum-based chemotherapy, multivariate analysis demonstrated that the FOLFOX regimen in comparison to the combination of 5-FU and cisplatin was associated with an improved median PFS, 6.9 versus 4.8 months (*p* < 0.0001), and an improved median OS, 17.8 versus 9.3 months (*p* = 0.02), respectively. For all first-line regimens, the median OS was 15.1 months [24].

These results are similar to those from three prospective phase II trials, in which encouraging objective response rate (ORR) and OS were reported for the combination of a fluoropyrimidine and oxaliplatin (FOLFOX or CAPOX) in the front-line setting. In these three trials, ORR ranged from 45 to 50%, and median OS ranged from 15.2 to 20.4 months [16,17,18] (Table 2). In these studies, toxicities were generally mild, with the most common grade III or IV treatment-related toxicities being hematological (neutropenia or thrombocytopenia) in 10% to 63%, fatigue in 3% to 30%, diarrhea in 3% to 10%, nausea or vomiting in 3% to 10%, and peripheral neuropathy in 9% to 25% of the patients.

The association of 5-FU, mitomycin, and doxorubicin was evaluated in one phase II multicenter study involving 39 patients with SBA or adenocarcinoma of the ampulla of Vater. This combination demonstrated disappointing results with an ORR of 18.4% and a median OS of 8 months [15]. As expected, the most frequent toxicities were hematological and gastrointestinal, with grade III to V toxicities reported in 26 (72%) of the 36 patients evaluated and with one patient dying from grade V hematological toxicity [15].

Another first-line option is triplet chemotherapy consisting of 5-FU, oxaliplatin, and irinotecan (CAPIRINOX or FOLFIRINOX) as it is utilized for metastatic colorectal or pancreatic cancer. However, in a recent phase II study involving 33 patients, McWilliams et al. reported a median OS of 13.4 months and a disease control rate of 37.5% in patients with advanced SBA with the first-line use of CAPIRINOX [20]. Although not formally compared, the response rate and survival for CAPIRINOX are lower than those described in the phase II trials with CAPOX or FOLFOX [16,17,18,20] (Table 2). 

### 2.2. Second-Line Chemotherapy Regimen 

Few data exist for second-line chemotherapy (Table 2). In the small series by Locher et al., second-line FOLFIRI demonstrated a median PFS of 5 months in eight patients. Among these eight patients, five had a clinical benefit [22]. These results are supported by a multicenter retrospective series reported by Zaanan et al., in which 28 patients treated with FOLFIRI demonstrated an ORR and disease control rate of 20% and 50%, respectively. In the same study, median PFS and OS were 3.5 months and 10.5 months, respectively [25]. In this series, 48% of the patients had grade III or IV therapy-related adverse events, mainly neutropenia in 37% and gastrointestinal in 14%. One treatment-related death due to grade V neutropenia was reported [25]. These data suggest a modest activity and a relatively manageable toxicity profile for FOLFIRI as a second-line treatment in patients with progressive advanced SBA after platinum-based first-line chemotherapy. 

Two other small studies suggested clinical activity for taxane-based therapy in advanced SBA after first-line chemotherapy [19,28]. In a retrospective study, Aldrich et al. reported the results from taxane-based therapy in 20 patients demonstrating an ORR of 65% and a median PFS and OS of 3.8 and 10.7 months, respectively [28]. A prospective phase II study reported by Overman et al. reported similar results for 10 evaluable patients with advanced SBA treated with nab-paclitaxel with median PFS and OS of 3.2 and 10.9 months, respectively [19]. Although the sample sizes are small, taxane chemotherapy might represent a novel therapeutic option for SBA patients; however, replication in larger datasets is needed [19,28].

Limited data exist regarding other chemotherapy regimens in advanced SBA. Some responses with gemcitabine were described in several studies, suggesting a benefit from this drug [9,23]. In the retrospective series by Fishman et al., an ORR of 33.3% for gemcitabine alone and an ORR of 50% for the combination of a fluoropyrimidine and gemcitabine in first or second-line setting was reported [8]. Another study reported no responses in the first-line therapy with gemcitabine but one response among two patients treated by gemcitabine in the second-line setting [23]. More recently, a retrospective study reported by Aydin et al. demonstrated an ORR of 20%, a median PFS of 6 months, and a median OS of 11 months among 10 patients treated by gemcitabine monotherapy in the first-line setting [27]. These data appear to support the clinical activity of gemcitabine in advanced SBA. 

## 3. Targeted Therapies

Due to some similarities between SBA and colorectal cancer (CRC) in terms of molecular alterations and effective chemotherapy regimens, targeted therapies usually used in CRC, such as anti-epidermal growth factor receptor (EGFR) or anti-vascular endothelial growth factor (VEGF), were studied in advanced SBA. 

### 3.1. Anti-Angiogenic Therapies 

The role of anti-angiogenic therapies, such as bevacizumab or ramucirumab, which were well established in metastatic CRC, underwent limited evaluation in advanced SBA, despite the known importance of the VEGF pathway in SBA [29]. Indeed, immunohistochemical expression of VEGF-A was recently reported as a potentially useful biomarker for the prediction of the efficacy of bevacizumab-based treatment in patients with advanced SBA [30].

Most studies evaluating bevacizumab in advanced SBA are retrospective and have suggested a survival benefit with the addition of bevacizumab to standard chemotherapy (Table 3) [31,32,33].

Only one phase II trial evaluated the combination of bevacizumab and CAPOX in metastatic SBA or ampullary adenocarcinoma. This study reported a 6 month PFS rate of 68% and an ORR of 48% [34]. However, despite the absence of statistical comparison, these results did not appear improved compared to similar results reported from an earlier conducted phase II trial of CAPOX [16].

An ongoing randomized phase II trial is currently evaluating ramucirumab and paclitaxel versus FOLFIRI in refractory SBA previously treated by a fluoropyrimidine and/or oxaliplatin. This trial may help define the therapeutic algorithm for second-line chemotherapy in advanced SBA (NCT 04205968) (Table 3).

### 3.2. Anti-EGFR 

By analogy with CRC, studies have evaluated the benefit of anti-EGFR antibodies in *RAS* wild-type advanced SBA [35,36,37,38] (Table 3). Indeed, the proportion of *RAS* mutations in SBA is around 40% to 53%, initially suggesting a theoretical potential efficiency of anti-EGFR antibodies may apply to approximately one-half of all SBA [43,44].

An early case series of four patients suggested a benefit from cetuximab when combined with irinotecan in patients with advanced SBA. Among these four patients, one patient obtained a complete response, two patients had a partial response, and one patient had stable disease at the first evaluation. Among the three patients tested for *KRAS* mutations, all had a wild-type mutational status and corresponded to the two partial responses and the one complete response [35]. 

Another retrospective multicenter study involving 13 patients with metastatic SBA receiving anti-EGFR antibody (cetuximab or panitumumab) in monotherapy or in association with chemotherapy in first- or second-line treatment reported a median PFS of 5.5 months, a median OS of 15.8 months, and a complete response rate of 15%, a partial response rate of 39%, stable disease rate of 23%, and progression disease rate of 15%. However, in this work, *RAS* mutational status was not reported [36].

By focusing on *KRAS* wild-type metastatic SBA patients, another retrospective single-center study reported on 25 patients treated with either cetuximab (*n* = 19) or panitumumab (*n* = 6) as a single agent or in combination with chemotherapy. In this study, an ORR of 12%, a DCR of 36%, and a progressive disease rate of 64% were reported [37].

The results of a recent single-arm phase II study also reported disappointing results with no responses seen from panitumumab monotherapy in nine metastatic *RAS* wild-type SBA and ampullary adenocarcinoma patients [38].

Apart from *RAS* mutations, *EGFR* mutations were reported in approximately 2.5% of SBA [44], leading to an evaluation of EGFR tyrosine kinase inhibitors, such as erlotinib in advanced SBA. In a case report of a patient with concomitant lung and duodenal adenocarcinoma with an *EGFR* mutation, the treatment of erlotinib and S-1 resulted in a partial response [39] (Table 3).

### 3.3. Other Promising Targeted Therapies

The recently improved understanding of the mutational landscape of SBA may lead to specific targeted therapies, such as anti-HER2 antibodies, anti-MEK tyrosine kinase, PIK3CA inhibitors, or NTRK-directed therapies. The use of anti-HER2 antibodies was reported in one case report in which aHER2 amplified duodenal cancer demonstrated a response to trastuzumab and FOLFOX in the neoadjuvant setting [45].

Though NTRK inhibitors have not been evaluated in advanced SBA, they already demonstrated anti-tumoral activity in a variety of tumor types with *NTRK* fusions [46]. It might therefore be administrated in case of this molecular alteration in advanced SBA [47].

## 4. Immune Checkpoint Inhibitors 

Immune checkpoint inhibitors (ICIs) appear today as the cornerstone of immunotherapy in several cancer types and are mainly represented by antibodies targeting anti-programmed cell death protein 1 (PD-1), anti-programmed cell death ligan-1 (PD-L1), and anti-cytotoxic T-lymphocyte antigen-4 (CTLA-4). Predictive biomarkers of ICIs, such as PD-L1 expression, combined positive score (CPS), microsatellite instability (MSI), and tumor mutational burden (TMB), are currently used in several types of cancers [48,49,50,51,52]. Predictive biomarkers for the efficiency of ICI in SBA remain under evaluation.

The phase II basket trial, KEYNOTE-158, investigated pembrolizumab, an anti-PD-1 antibody, in advanced MSI solid tumors that experienced failure with prior therapy. For the subgroup of 19 patients with advanced SBA, the reported ORR was 42.1%, median PFS was 9.2 months, and median OS was unreached [40]. 

In the recent ZEBRA multicenter phase II study involving 40 patients with unresectable or metastatic SBA, pembrolizumab was evaluated as second-line treatment. In this study, 50% of the four patients with MSI-high tumors had a partial response, while 3% of the 32 MSS patients had a partial response. MSI status was unknown for four patients in this study [41].

Another single-agent, open-label, phase II study evaluating avelumab, an anti-PD-L1 antibody, in patients with advanced or metastatic SBA reported that avelumab was considered safe, and antitumor activity was observed as evidenced by a 29% RR and 71% DCR [42].

Combinations of ICIs together or ICIs combined with targeted therapies or chemotherapy are also being investigated. Two phase II studies are currently ongoing for the treatment of rare tumors, and both include SBA. The first trial combines ipilimumab (an anti-CTLA-4 antibody) with nivolumab (an anti-PD-1 antibody) compared to nivolumab monotherapy (NCT 02834013). The second trial compares cobimetinib (a mitogen-activated protein kinase (MEK) inhibitor) and atezolizumab (an anti-PD-L1 antibody) (NCT03108131). The results of targeted therapies of ICIs investigated in advanced SBA are summarized in Table 3.

A major challenge for SBA is the need to develop biomarkers to predict clinical benefits from ICIs. Apart from biomarkers such as MSI status, tumor mutational burden, and tumor-infiltrating lymphocytes, other potential biomarkers of interest should be explored. For example, the gut microbiome could have crosstalk with cancer immune response and immunotherapy [53]. Furthermore, ferroptosis, a form of regulated cell death mainly relying on iron-mediated oxidative damage and subsequent cell membrane damage, seems to affect the efficacy of cancer treatments, and thus combinations with agents targeting ferroptosis signaling may be of relevance to SBA [54]. However, there is currently no specific data evaluating the gut microbiome or ferroptosis with regard to tumor response in the treatment of SBA.

## 5. Specific Cases Management in Advanced SBA

### 5.1. Resection of Small Bowel Adenocarcinoma Metastases

There are limited data regarding the prognosis of patients with resected metastatic SBA. Therefore, outcomes of a sub-group of 34 patients with curatively resected metastatic SBA of the ARCAD-NADEGE study were analyzed. The metastatic sites were mainly peritoneal (29.4%), liver (26.5%), lymph nodes (11.8%), lung (2.9%), multiple (14.7%), or other (14.7%) [55]. The median OS for these patients undergoing curative-intent metastatic resection of SBA was 28.6 months, which is better than the median OS of 12.7 months for all metastatic patients in the NADEGE cohort. Although there are major limitations to cross-trial comparisons, this OS compares favorably to those recently reported in studies of patients treated with palliative chemotherapies [15,16,17,18,19,20]. However, patients with resected metastasis in the ARCAD-NADEGE cohort were highly selected patients with 85% (*n* = 29) of patients with a solitary metastatic site and therefore not representative of all metastatic SBA patients. Among all metastatic SBA resected patients, 30 (88.2%) also received perioperative chemotherapy, though perioperative chemotherapy was not associated with a better OS in this subgroup of patients [55]. Negative predictive factors for OS after metastasectomy were poor differentiation, positive margins, and lymphatic invasion. 

### 5.2. Peritoneal Carcinomatosis

Peritoneal carcinomatosis is a frequent site of disease ranging from around 25 to 50% of metastatic SBA [6,55,56]. Hyperthermic intraperitoneal chemotherapy (HIPEC) combined with cytoreductive surgery (CRS) was evaluated for SBA patients with peritoneal carcinomatosis. In eight observational studies, survival outcomes and toxicities were evaluated [57,58,59,60,61,62,63] (Table 4). In these studies, median OS from CRS + HIPEC ranges from 16 to 47 months, and grade III or IV treatment-related toxicities range from 12 to 35% (Table 4). The main observed complications are post-operative infections, abdominal collections, hematological toxicity, re-interventions, and pleural effusions. 

In the largest multicenter study by Liu et al. with 152 patients receiving CRS + HIPEC between 1989 and 2016, the median OS was 32 months with a median disease-free survival of 14 months. In the multivariate analysis, a Peritoneal Cancer Index (PCI) ≤ 15 was independently associated with an improvement in OS (*p =* 0.003) [62]. However, despite a tolerable rate of grade III or IV treatment-related toxicities (19.1%) there was a 2% death rate due to multiorgan failure.

Thus, in SBA patients with limited peritoneal carcinomatosis (PCI ≤ 15) and a physical status allowing for a major surgical procedure, CRS and HIPEC could be considered in expert centers [57,59,60,61,62,64,65].

## 6. Conclusions

Systemic chemotherapy appears to be benefit patients with advanced SBA compared to best supportive care alone, though data from randomized controlled trials are lacking.

The combination of a fluoropyrimidine and oxaliplatin (FOLFOX or CAPOX) remains the most used and effective first-line chemotherapy with manageable toxicities. Alternatives in the first-line are 5-FU and cisplatin or FOLFIRI. In the second-line setting, FOLFIRI appears to be an effective option. Taxane-based therapy also appears to be clinically active, but further evaluation in larger series is needed. 

The benefit of targeted therapies for SBA is uncertain and continues to be investigated. Immune checkpoint inhibitors have demonstrated robust activity for the subset of SBA with MSI. Due to the rareness of the disease, inclusion in clinical trials should be prioritized when feasible. 

For selected patients, surgical resection of metastatic disease can be considered, especially in cases of isolated peritoneal, liver, or lung metastasis. For the specific location of peritoneal carcinomatosis, SBA patients with limited peritoneal disease should be considered for HIPEC and CRS at centers of excellence.

## Figures and Tables

**Table 1 cancers-14-01137-t001:** Retrospective and prospective studies comparing chemotherapy versus best supportive care in advanced small bowel adenocarcinoma.

References	Type of Study	Number of Patients			Type of CT	Number of Patients (%)	Patients Treated with CT (Months)			OS in Patients without CT (Months)	*p*-Value
		Total	Receiving CT	Not Receiving CT			ORR	DCR	OS (Months)		
Dabaja, Cancer, 2004 [6]	Retrospective	75	34	41	NA	-	-	-	12	2	0.02
Fishman, Am J Oncol, 2006 [9]	Retrospective	105	44	61	Various regimens	-	29.5%	50%	19	13	0.035 (in multivariate analysis)
Moon, J Cancer Res Clin Oncol, 2010 [11]	Retrospective	87	34	53	All	-	27.6%	-	9	3.5	0.01
					5-FU	4 (11.8%)					
					5-FU–cisplatin	11 (32.4%)					
					5-FU-based adriamycin	14 (41.2%)					
					Gemcitabine	2 (5.8%)					
					Other regimens	3 (8.8%)					
Czaykowski, Clin Oncol, 2007 [10]	Retrospective	37	16	21	5-FU based		5%	-	15.6	7.7	NA
Koo, BMC Cancer, 2011 [12]	Retrospective	91	40	51	All	-	11.1%	37.0%	11.8	4.1	<0.01
					5-FU	10 (24.0%)					
					5-FU–cisplatin	25 (62.5%)					
					FOLFIRI	3 (7.5%)					
					5-FU–adriamycin–mitomycin	2 (5.0%)					
Khan, BMC Cancer, 2015 [13]	Retrospective	59	46	59	Various regimens	-	50%	-	60.9% (1-year OS)	27.1% (1-year OS)	0.04
Halfdanarson, Am J Surg, 2010 [14]	Retrospective	165	NA	NA	NA	-	-	-	15.5	5.3	<0.01
Aparicio, Int J Cancer, 2020 [4]	Prospective	124 *	86	15	All	-	-	-	14.6	2.2	0.0001
					5-FU	6 (7.0%)	-	-	-	-	-
					FOLFOX/CAPOX	60 (69.8%)	-	-	-	-	-
					FOFLIRI	16 (18.6%)	-	-	-	-	-
					Other regimens	4 (4.6%)	-	-	-	-	-

Abbreviations: DCR: disease control rate; CT: chemotherapy; NA: not available; ORR: objective response rate; OS: overall survival * 23 patients with NA data.

**Table 2 cancers-14-01137-t002:** Evaluation of different chemotherapy regimens in advanced small bowel adenocarcinoma.

References	Type of Study	Number of Patients	Line of Treatment	Type of CT (Number of Patients)	ORR	Disease Control Rate	Median PFS (Months)	Median OS (Months)
Crawley, Br J Cancer, 1998 [21]	Retrospective Single Center	8	1	All (*n* = 8)	-	75%	7.8	13
				ECF (*n* = 5)	-	80%	-	-
				5-FU (*n* = 2)	-	50%	-	-
				5-FU–mitomycin (*n* = 1)	-	100%	-	-
Locher, Oncology, 2005 [22]	Retrospective Single Center	20	1 2	All (*n* = 20)	21%	79%	8	14
				5-FU–cisplatin (*n* = 15)	20%	-	-	-
				5-FU–carboplatin (*n* = 2)	0%	-	-	-
				FOLFOX (*n* = 3)	33%	-	-	-
				All (*n* = 13)	-	-	-	-
				FOLFIRI (*n* = 8)	-	50%	5	-
				5-FU–cisplatin (*n* = 1)	0%	0%	-	-
				5-FU (*n* = 4)	0%	0%	-	-
Overman, Cancer, 2008 [23]	Retrospective Single Center	80	1	All (*n* = 80)	71%	-	4.6	13.0
				5-FU–cisplatin (*n* = 29)	41%	-	8.7	14.8
				5-FU alone/non 5-FU-based (*n* = 51)	30%	-	3.9	12.0
Zaanan, Ann Oncol, 2010 [24]	Retrospective Multicentric	93	1	All (*n* = 93)	26%	74%	6.6	15.1
				5-FU (*n* = 10)	0%	50%	7.7	13.5
				FOLFOX (*n* = 48)	34%	79%	6.9	17.8
				FOLFIRI (*n* = 19)	34%	73%	6.0	10.6
				5-FU-cisplatin (*n* = 16)	31%	69%	4.8	9.3
Zaanan, Cancer, 2011 [25]	Retrospective Multicentric	28	2	FOLFIRI (*n* = 28)	20%	52%	3.2	10.5
Tsushima, Oncologist, 2012 [26]	Retrospective Multicentric	132	1	All (*n* = 132)	-	-	-	-
				5-FU (*n* = 60)	20%	-	5.4	13.9
				5-FU–cisplatin (*n* = 17)	38%	-	3.8	12.6
				FOLFOX (*n* = 22)	42%	-	8.2	22.2
				FOLFIRI (*n* = 11)	25%	-	5.6	9.4
				Other regimens (*n* = 22)	21%	-	3.4	8.1
Aydin, J BUON, 2016 [27]	Retrospective Multicentric	56	1	All (*n* = 56)	-	-	7	13
				FOLFOX (*n* = 18)	35%	-	7	15
				5-FU–cisplatin (*n* = 17)	56%	-	8	11
				FOLFIRI (*n* = 18)	55%	-	8	16
				Gemcitabine (*n* = 10)	20%	-	5	11
Aldrich, Oncologist, 2019 [28]	Retrospective Single Center	20	1 or 2	All: Taxane-based (*n* = 20)	65%	-	3.8	10.7
				Combination (*n* = 17)	-	-	-	-
				Monotherapy (*n* = 3)	-	-	-	-
Gibson, Oncologist, 2005 [15]	Phase II Multicentric	39	1	5-FU–mitomycin–doxorubicin	18.4%	31%	5	8
Overman, J Clin Oncol, 2009 [16]	Phase II Single Center	30	1	CAPOX	50%	87%	11.3	20.4
Xiang, Anticancer Drugs, 2012 [17]	Phase II Multicentric	33	1	FOLFOX	48.5%	-	7.8	15.2
Horimatsu, Int J Clin Oncol, 2017 [18]	Phase II Multicentric	24	1	FOLFOX	45%	80%	5.4	17.3
Mc Williams, Cancers, 2017 [20]	Phase II Single Center	32	1	CAPIRINOX	37.5%	-	8.9	13.4
Overman, Ann Oncol, 2018 [19]	Phase II Single Center	10	>1	Nab-paclitaxel	20%	50%	3.2	10.9

Abbreviations: ORR: objective response rate; OS: overall response; PFS: progression-free survival; CT, chemotherapy.

**Table 3 cancers-14-01137-t003:** Main studies evaluating targeted therapies or immune checkpoint inhibitors in advanced small bowel adenocarcinoma.

References	Type of Study	Number of Patients	Investigated Therapies	Main Results
Aydin, Clin Colorectal Cancer 2017 [31]	Retrospective Multicenter	28	Bevacizumab + CT (*n* = 12) CT alone (*n* = 16)	ORR: 43.7% in the CT group vs. 58.3% in the bevacizumab + CT group (NS) mPFS: 7.7 months in the CT group vs. 9.6 months in the bevacizumab + CT group (NS) mOS: 14.8 months in the CT group vs. 18.5 months in the bevacizumab + CT group (NS)
Takayoshi, Cancer Chemother Pharmacol 2017 [32]	Retrospective Single Center	33	Bevacizumab + CT (*n* = 9) CT + other targeted therapy or CT alone (*n* = 24)	ORR: 33%.3 in the bevacizumab + CT group DCR: 55.5% in the bevacizumab + CT group mOS: 11.4 months among the 24 patients who did not receive bevacizumab in any of the treatment lines versus 21.9 months in the bevacizumab + CT group (*p* = 0.179).
Hirao, Oncol Lett, 2017 [33]	Retrospective Single Center	17	Bevacizumab + CT (*n* = 7) Cetuximab + CT (*n* = 1) CT alone (*n* = 9)	Among the 17 patients who received oxaliplatin-based CT in first-line, a PS of 0 (*p* = 0.0255) and treatment with bevacizumab (*p* = 0.0121) were significant positive prognostic factors.
Gulhati, Cancer, 2017 [34]	Phase II Single Center	30	Bevacizumab + CT (CAPOX)	ORR: 48.3% (1 complete response, 13 partial responses) 6 months PFS: 68%–mPFS: 8.7 months–mOS: 12.9 months
NCT04205968	Phase II	-	Ramucirumab + CT (paclitaxel) versus CT (FOLFIRI)	Ongoing
Santini, Br J Cancer 2010 [35]	Retrospective Single Center	4	Cetuximab + CT (irinotecan)	ORR: 75%–DCR: 100% All 3 responders were KRAS wild-type
Dell’Aquila, Clin Med Insights Oncol, 2020 [36]	Retrospective Multicenter	13	Cetuximab + CT	ORR: 44% (CR in 15% of patients, PR in 39% of patients) DCR: 67%
Serpas, J Clin Oncol, 2018 [37]	Retrospective Single Center	25	Cetuximab ± CT (*n* = 19) Panitumumab ± CT (*n* = 6)	mPFS: 3.3 months–mOS: 13 months ORR: 12%–DCR 36%. All 3 responders were treated in combination with irinotecan and did not previously progress on irinotecan Patients treated by anti-EGFR in monotherapy (*n* = 7): mPFS: 2.2 months ORR: 0% and best response: SD (*n* = 1)
Gulhati, Oncologist, 2018 [38]	Phase II	9	Panitumumab monotherapy	ORR: 0%–DCR: 22% (*n* = 2) mPFS: 2.4 months–mOS: 5.7 months No patients had extended RAS mutations (exons 2/3/4), 2 had a BRAF G469A mutation, and 1 had a PIK3CA H1074R mutation.
Akiyama, Intern Med, 2017 [39]	Case report	1	Erlotinib + S-1	Concomitant response of lung and duodenal cancer to the combination of erlotinib and S-1
Marabelle, J Clin Oncol, 2020 [40]	Phase II	19 SBA among 233 patients	Pembrolizumab monotherapy	ORR: 34.3% mPFS: 4.1 months–mOS: 23.5 months
Pedersen, Clin Cancer Res, 2021 [41]	Phase II	40	Pembrolizumab monotherapy	PR: 8% (*n* = 3)–Unachieved predefined success criteria of ORR 30%. mPFS: 2.9 months–mOS: 7.1 months 50% of patients with MSI-H tumors achieved PR and remained alive without progression 1 confirmed PR (3%) was seen in patients with low MSS/MSI tumors and correlated with high tumor mutation burden.
Cardin, J Clin Oncol, 2O2O [42]	Phase II	-	Avelumab	29% RR and 71% DCR
NCT02834013	Phase II	-	Ipilimumab + nivolumab versus nivolumab	Ongoing (dedicated to rare tumors, including SBA)
NCT03108131	Phase II	-	Cobimetinib + atezolizumab	Ongoing (dedicated to rare tumors, including SBA)

Abbreviations: CT: chemotherapy; DCR: disease control rate; MSI-H: Microsatellite instability high; NS: non-significant; ORR: objective response rate; mOS: median overall survival; mPFS: median progression-free survival; CR: complete response; PR: Partial response; SD: stable disease.

**Table 4 cancers-14-01137-t004:** Main observational studies evaluating intraperitoneal chemotherapy and cytoreductive surgery.

References	Type of Study	Number of Patients	Median OS (Months) from CRS + HIPEC	Median OS (Months) from Diagnosis	Grade III–IV Toxicities	Treatment-Related Death
Marchettini, Eur J Surg Oncol, 2002 [57]	Retrospective Single Center	6	16	-	-	0%
Jacks, J Surg Oncol, 2005 [58]	Retrospective Single Center	6	30.1	54	-	-
Chua, J Surg Oncol, 2009 [59]	Retrospective Single Center	7	25	-	29%	0%
Elias, Ann Surg, 2010 [60]	Retrospective Multicenter	31	47	-	35%	2.9%
Sun, Am Surg, 2013 [64]	Prospective Single Center	17	18	37	12%	0%
van Oudheusden, Surgery, 2015 [61]	Retrospective Multicenter	16	31	-	25%	NA
Liu, Ann Surg Oncol, 2016 [63]	Retrospective Single Center	31	36	51	25.8%	0%
Liu, Ann Surg, 2018 [62]	Retrospective Multicenter	152	32	-	19.1%	2%

Abbreviations: OS: overall survival; CRS: cytoreductive surgery; HIPEC: hyperthermic intraperitoneal chemotherapy.

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
