# Peer review of "Therapeutic Strategies for Patients with Advanced Small Bowel Adenocarcinoma: Current Knowledge and Perspectives"

_cancers, 2022, doi:10.3390/cancers14051137_

Round 1
Reviewer 1 Report
The manuscript by Moati and Zaanan provides a comprehensive review of the literature that investigate clinical treatment of the patients with advanced small bowl adenocarcinoma (SBA). The authors have done a wonderful job on collating the related reports and on discussing the advantages, weaknesses, and perspectives of currently available therapeutic options for treating SBA. This review would be of great interests for the people who are involved in treating or studying SBA, perhaps also other aggressive cancer types originating from epithelial abnormalities. I will recommend acceptance of this manuscript for publication, however, improvement may be made on the following points.
- The introduction needs to be expanded a little. Delineate more about SBA definition, diagnosis, prognosis and how the disease impacts patients’ life physically and economically.
- Provide more summarizing description to the Tables (1-4) in the text so that readers can have a quick idea of what was occurring with those therapeutic options.
- Some sentences do not read well. A few sentences are too long and may need to break up. Some words are not appropriately used. For example, a) Line 38, “ survival (OS) rate is higher to 50%” is confusing; b) Line 77, the word “resumed” ? C) “There is to report one treatment-related death…” may be changed into “One treatment-related death was reported with a patient having grade V neutropenia.”; d) “study among (number) patients” (in a few sentences) may be changed into “study of (or involving, etc.); e) The sentence in Line 160-161 reads awkward, “treated by” should be “treated with”; f), Grammar errors in the sentence in Line 172-174, “has” is “have”, “despite the important proportion of these …” needs to corrected; g), Line 196, “a wild-type mutational status” is not favorable description. The authors may need to check for more of these throughout the manuscript.
Author Response
Dear reviewer 1,
We respectfully thank you for your time and advices relative to our manuscript:
- The introduction has now been expanded.
- We tried to add few details in the text of results presented in the Table 1-4 for a better understanding of the different therapeutic options.
- The different inappropriate words have now been changed and the entire manuscript has been revised to correct English language.
We hope that this new version of the article will suit you.
Best regards,
Reviewer 2 Report
Moati and Zaanan in their review manuscript entitled “Therapeutic strategy for patients with advanced small bowel adenocarcinoma: current Knowledge and Perspectives” elegantly discuss the prognostic and therapeutic aspects of SBA with detailed (with ample citation of published research works) emphasis on the systemic as well as targeted chemotherapeutic approaches. In addition, the authors also included sections on immunotherapy, and a few specific case managements.
This reviewer thinks the authors could extend their discussion a little further by including two trending topics with enormous therapeutic potential, such as
- Gut microbiome and microbial metabolites
- Ferroptosis
Author Response
Dear reviewer 2,
We respectfully thank you for your time and advices relative to our manuscript.
In response to your comments, we completed our discussion in adding references to gut microbiome and ferroptosis in cancer response to treatment.
We hope that this new version of the article will suit you.
Best regards,
Reviewer 3 Report
This is an interesting review article regarding update management of small bowel adenocarcinoma. The followings are my comments.
#1. As small bowel adenocarcinoma, do the authors consider to divide the "small bowel" into duodenum, jejunum or jejunum ? Is there any effect of anatomical distribution of SBA and the clinical outcome ?
#2. SBA in patient with Crohn's disease (Int J Colorectal Dis. 2022 Jan;37(1):239 ) is a special sub-population. The authors may add review in this population.
#3. Line 175, author may add recent study from Japan ( BMC Cancer, 2021 Aug 31;21(1):978 ) the discuss possible use of biomarker in choosing the chemotherapy regimen for SBA.
Author Response
Dear reviewer 3,
We respectfully thank you for your time and advices relative to our manuscript:
- We thank you for asking about this precision. Despite a negative prognostic impact of the duodenal localization in SBA, tumor location had no impact on the response to chemotherapy. We now integrated this notion in the manuscript.
- We tried to mention this particular subgroup of interest in the introduction but according to our research, there was no data about a correlation between the presence of Crohn’s disease and the response to systemic treatment.
- We kindly thank you for your relevant suggestion. The corresponding reference relative to the correlation between VEGFA expression and bevacizumab efficiency in advanced SBA has now been added.
We hope that this new version of the article will suit you.
Best regards,
Round 2
Reviewer 3 Report
Dear Editor
The authors response to the questions raised and I have no more questions.
Author Response
Dear reviewer,
We thank you very much for your return.
Best regards,